# Bio-Potency and Molecular Docking Studies of Isolated Compounds from *Grewia optiva* J.R. Drumm. ex Burret

**DOI:** 10.3390/molecules26072019

**Published:** 2021-04-01

**Authors:** Wasim Ul Bari, Najeeb Ur Rehman, Ajmal Khan, Sobia Ahsan Halim, Ye Yuan, Mark A.T. Blaskovich, Zyta M. Ziora, Muhammad Zahoor, Sumaira Naz, Riaz Ullah, Amal Alotaibi, Ahmed Al-Harrasi

**Affiliations:** 1Department of Chemistry, University of Malakand, Chakdara, Dir Lower 18800, KPK, Pakistan; wasimouch4080@gmail.com; 2Natural and Medical Sciences Research Center, University of Nizwa, P.O. Box 33, Birkat Al Mauz 616, Nizwa, Oman; najeeb@unizwa.edu.om (N.U.R.); ajmalchemist@yahoo.com (A.K.); Sobia_halim@unizwa.edu.com (S.A.H.); aharrasi@unizwa.edu.com (A.A.-H.); 3Centre for Superbug Solutions, Institute for Molecular Bioscience, The University of Queensland, Brisbane, QLD 4072, Australia; ye.yuan1@uq.net.au (Y.Y.); m.blaskovich@imb.uq.edu.au (M.A.T.B.); z.ziora@uq.edu.au (Z.M.Z.); 4Department of Biochemistry, University of Malakand, Chakdara, Dir Lower 18800, KPK, Pakistan; sumaira.biochem@gmail.com; 5Department of Pharmacognosy (MAPPRC), College of Pharmacy, King Saud University, Riyadh 11451, Saudi Arabia; rullah@ksu.edu.sa; 6Basic Science Department, College of Medicine, Princess Nourah Bint Abdulrahman University, Riyadh 11671, Saudi Arabia

**Keywords:** *Grewia optiva*, antioxidant activity, molecular docking, alpha-amylase, anticholinesterase activity, chemical composition

## Abstract

In the study, two novel compounds along with two new compounds were isolated from *Grewia optiva*. The novel compounds have never been reported in any plant source, whereas the new compounds are reported for the first time from the studied plant. The four compounds were characterized as: 5,5,7,7,11,13-hexamethyl-2-(5-methylhexyl)icosahydro-1H-cyclopenta[a]chrysen-9-ol (**IX**), docosanoic acid (**X**), methanetriol mano formate (**XI**) and 2,2’-(1,4-phenylene)bis(3-methylbutanoic acid **(XII)**. The anticholinesterase, antidiabetic, and antioxidant potentials of these compounds were determined using standard protocols. All the isolated compounds exhibited a moderate-to-good degree of activity against acetylcholinesterases (AChE) and butyrylcholinesterase (BChE). However, compound **XII** was particularly effective with IC_50_ of 55 μg/mL (against AChE) and 60 μg/mL (against BChE), and this inhibitory activity is supported by in silico docking studies. The same compound was also effective against DPPH (2,2-diphenyl-1-picrylhydrazyl) and ABTS (2,2′-azinobis-3-ethylbenzothiazoline-6-sulfonic acid) radicals with IC_50_ values of 60 and 62 μg/mL, respectively. The compound also significantly inhibited the activities of α-amylase and α-glucosidase in vitro. The IC_50_ values for inhibition of the two enzymes were recorded as 90 and 92 μg/mL, respectively. The in vitro potentials of compound **XII** to treat Alzheimer’s disease (in terms of AchE and BChE inhibition), diabetes (in terms of α-amylase and α-glucosidase inhibition), and oxidative stress (in terms of free radical scavenging) suggest further in vivo investigations of the compound for assessing its efficacy, safety profile, and other parameters to proclaim the compound as a potential drug candidate.

## 1. Introduction

There are about 159 species of *Grewia,* found generally in tropical and subtropical areas. The majority of the species are from China, India, Pakistan, Bangladesh, Malaysia, Madagascar, northern Thailand, Australia, and South Africa. In Pakistan, 10 species have been identified; these are often used in traditional medicine to cure cough, smallpox, fever, diarrhea, malaria, dysentery, eczema, and typhoid. This traditional use is increasingly supported by recent scientific research—some species of this genus have now been identified to have antibacterial, antioxidant, antimalarial, antidiabetic, and memory enhancement properties [1,2]. A number of chemical compounds such as nitidanin, grewin, harman, and alkaloid-containing lignans, have been isolated from different species of the genus *Grewia*. Some important glucosides, including gulonic acid, contanoic acid, and vitexin, have also been isolated from this genus [3]. Joshi et al. isolated seven compounds from the roots of *G. microcos*: N-methyl-6-β-(1′,3′,5′-trienyl)-3-β-methoxyl-3-β-methylpiperidine, dioctyl phthalate, octadecadienoic acid, dihydroxy-3-propenchalcone, stigmasterol, dibutyl phthalate, and ursolic acid [4]. Among these compounds, stigmasterol is antibacterial, while ursolic acid is anti-inflammatory and antihyperlipidemic agent that is also used for the enhancement of defense systems. It also have antitumor potential while the isolated cyclopentadeca-4, 12-dienone from the extract of *Grewia hirsute* has been found to have an anti-diabetic potential [5]. Hiba et al. have reported the isolation of octanoic acid, tridecanoic acid, octadecatrienoic acid, and eicosanoic acid from *Grewia tenax* [6]. Additionally, previously, we evaluated the *Grewia optiva* extracts for different biological potentials and isolated eight compounds from this plant [7,8,9].

Natural products are known to have anti-diabetic effects and offered plentiful exciting potentials for the future development and improvement of successful therapies [10]. Interestingly, previously isolated bioactive components from medicinal plants, endophytes, marine species, and oleo-gum resins demonstrated promising α-glucosidase activity [11,12,13,14,15,16]. Diabetes mellitus (DM) is one of the most common and serious metabolic diseases characterized by high blood glucose levels (hyperglycemia), and their complications increase the morbidity and mortality threats for Type 2 diabetes patients [17]. According to World Health Organization (WHO) assessments, approximately 90% of the world’s diabetic people have type 2 diabetes mellitus, and from 2012 to 2014, about 1.5 million peoples died from complications of this disease [11,12,13,14,15]. Clinically approved anti-diabetic drugs α-glucosidase inhibitors (AGIs) have restricted safety alarms, temporally recover the blood glucose levels, and improve type 2 DM (diabetes mellitus) complications, together with the treatment of obesity [12,13]; however, these AGIs are known to cause flatulence, diarrhea, and abdominal discomfort [15]. Due to the crucial role of this enzyme in hyperglycemia and the side effects of the existing synthetic drugs, there is an urgent need to discover safe and effective enzyme inhibitors as an approach to effectively control diabetic disorders.

Antioxidants are chemical substances that preclude oxidation or reduce the levels of free radicals in human bodies. In industries associated with food, synthetic antioxidants are used as preservatives, some of which historically have shown toxic or carcinogenic effects [17]. The search for natural antioxidants present in medicinal plants is very important due to their potential for extensive use in practical applications [17,18,19,20,21]. Additionally, natural compounds have been found to be promising in treating some neurodegenerative diseases [9]. For instance, two clinically approved drugs for the treatment of Alzheimer’s disease (AD), i.e., rivastigmine and galantamine, are derived from plant sources. Rivastigmine is a semi-synthetic derivative of physostigmine, while galantamine is an alkaloid originally isolated from the bulbs and flowers of a number of plants. Both of them are inhibitors of cholinesterases used to treat Alzheimer’s disease. Acetylcholine (ACh) and butyrylcholine (BCh) are important neurotransmitters in the procurement and storage of transmitted memory and are involved in the transmission of impulses across the synapse. Their levels are reduced in AD [7,8,9]. To restore and control the activities of ACh and BCh at the synapse, inhibitors of their respective cholinesterases, acetylcholinesterase (AChE), and butyrylcholinesterase (BChE) are used [7,8,9].

In the current study, we have isolated methanetriol mano formate (**XI**) and 2,2’-(1,4-phenylene)bis(3-methylbutanoic acid (**XII**), which are reported as natural products for the first time, while 5,5,7,7,11,13-hexamethyl-2-(5-methylhexyl)icosahydro-1H-cyclopenta[a]chrysen-9-ol (**IX**) and docosanoic acid (**X**) were reported for the first time from selected plant although they have been reported from this genus previously. All compounds were tested for their antidiabetic, antioxidant, and anticholinesterase potential.

## 2. Results

### 2.1. Spectroscopic Analysis of Isolated Compounds

The chemical structures of isolated compounds are presented in Figure 1.

Compound **IX:** (5,5,7,7,11,13-hexamethyl-2-(5-methylhexyl)icosahydro-1H-cyclopentachrysen-9-ol) was an amorphous solid with melting point of 210–225 °C. The proton and carbon-13 NMR (nuclear magnetic resonance) of this compound are presented in Appendix A. LC-MS (liquid chromatography-mass spectrometry: Appendix A) confirmed its molecular formula C_34_H_60_O and molecular mass, giving a molecular ion peak [M + H] at 484.85 g/mol. ^1^H NMR (600 MHz, DMSO-d6): δ0.80 (s, 12H), 0.89 (s, 6H,), 0.99 (d, J = 6.0 Hz 6H), 1.09 (m, 3H, H-7a,11a,13a), 1.39 (m, 6H, H-6,5b,111b,5a,3a,13b), 1.43 (m, 18H, H-10, 8, 11, 6, 13, 12, 4, 3, 2, 1), 1.57 (m, 1H, H-5‘), 1.31 (m 4H, H-3‘,4‘), 1.51 (m, 2H, H-2‘), 1.65 (m, 2H, H-1‘), 2.12 (m, 2H, H-8), 2.19(m, 1H, H-3a), 2.93 (m, 1H, H-9), 3.92(m, 2H, H-10), 6.54 (s, 1OH, H-9); ^13^C NMR (151 MHz, DMSO-d6): δ 16.65 (C-6‘), 18.59 (C-1), 19.52 (C-3‘), 19.94 (C-2‘), 23.54 (C-11,5‘), 29.91 (C-8,5b), 30.61 (C-5), 31.25 (C-13), 32.03 (C-1‘), 134.11 (C-2,7), 36.83 (C3,7a), 38.45(C-11b), 41.55(C-3a), 43.02 (C-4‘), 43.19(C-86,13a), 44.29(C-10), 47.01 (C-4), 48.86 (C-11a), 49.03 (C-13b), 55.88(C-5a), 56.38 (C-7a), 83.73 (C-9).

Compound **X:** (docosanoic acid): Docosanoic acid was isolated as a waxy solid, melting point is 81–85 °C, with molecular formula C_22_H_44_O_2_ and molecular mass of 340.592 g/mol. The spectral data (Appendix A ) were consistent with previously reported data. ^1^H NMR (600 MHz, Chloroform-d): δ 0.90 (t, J = 7.0 Hz, 3H, H-22), 1.28 (s, 36H, H-4-21), 1.66 (p, J = 7.5 Hz, 3H, H-3), 2.37 (t, J = 7.5 Hz, 2H, H-2), 10.20 (s, J = 1H, H-1); ^13^C NMR (151 MHz, Chloroform-d): δ 14.13 (C-22), 22.7 (C-21), 24.7 (C-3), 29.3 (C-4,5), 29.6 (C-6-19), 33.7 (C-2), 178.3 (C-1).

Compound **XI:** (methanetriol mano formate): Methanetriol mano formate was isolated for the first time as a natural product, having molecular formula C_2_H_3_O_4_ and ESI-HRMS (high resolution electrospray ionization mass spectrometry) *m/z* = 230.9633 [2M + 2Na + H]. The spectra of this compound have been provided in Appendix A. This compound has previously been prepared synthetically (Andersen and Carter, 2003). ^1^H NMR: δ 5.64 (s, 2OH), 8.92 (s, 1H); ^13^C NMR: δ 94.0 (C-2), 158.9 (C-1).

Compound **XII:** (2,2’-(1,4-phenylene)bis(3-methylbutanoic acid). Its spectra detail has been provided in Appendix A. ^1^H NMR (600 MHz, DMSO-d6): δ0.63 (s, 6H, H-4,5), 0.99 (s, 6H, H-4‘,5‘), 2.15 (m, 2H, H-3, 3‘), 3.15 (d, *J* = 10.3 Hz, 2H, H-2,2‘), 7.36 (m 4H, H-2‘‘, 3‘‘, 5‘‘, 6‘‘); ^13^C NMR (151 MHz, DMSO-d6): δ 20.19 (C-4,4,4‘,5‘), 31.57 (C-3,3‘), 58.89 (C-2,2‘), 128.78 (C-2‘‘,3‘‘,5‘‘,6‘‘), 132.12 (C-1‘‘,4‘‘), 174.88 (C-1,1‘).

### 2.2. Antioxidant Potential of Isolated Compounds

The antioxidant capabilities exhibited by the compounds are represented in Table 1. Amongst them, compound **XII** showed the highest scavenging with an IC_50_ value of 60 μg/mL with 86.54 ± 2.16 percent inhibition (at 1000 μg/mL) against 2,2-diphenyl-1-picrylhydrazyl (DPPH), followed by compound **IX**) having an IC_50_ value of 76 μg/mL with 80.21 ± 040 percent inhibition at 1000 μg/mL (Table 1). The most potent inhibitor of 2,2’-Azino-bis(3-ethylbenzthiazoline-6-sulfonic acid) (ABTS) free radicals was again compound **XII**, exhibiting (86.07 ± 1.43 inhibition at 1000 μg/mL and IC_50_ = 62 μg/mL. The values were compared with that of ascorbic acid (positive control) for which it was recorded as 35 μg/mL.

### 2.3. Inhibition of Cholinesterases

Effect on the activity of cholinesterases assessed in terms of IC_50_ is presented in Table 2. Compounds **XII** and **XI** effectively inhibited acetylcholinesterase (AChE), as indicated by their exhibited IC_50_ values calculated at 1000 μg/mL, 55 and 75 μg/mL, respectively, against AChE and 60 and 75, respectively, against butyrylcholinesterase (BChE). Compound **IX** also showed good % inhibition 78.11 ± 0.61 with an IC_50_ of 90 μg/mL against the same enzyme.

The values were compared with galantamine, which was used as positive control and showed an IC_50_ of 40 μg/mL against both enzymes. Molecular docking studies (see Section 2.5) support the binding of compound **XII** with these enzymes.

### 2.4. Effect of Isolated Compounds on the Activity of α-Glucosidase and α-Amylase

Effect on α-glucosidase’s activity is presented in Table 3. Compound **XII** had the highest inhibition potential amongst the compounds with an IC_50_ of 90 and % inhibition as 81.14 ± 1.06 at 1000 μg/mL, followed by compound **XI** with IC_50_ value = 100 μg/mL. Acarbose was used as a positive control, with an IC_50_ of 70 μg/mL.

The inhibition of α-amylase by these compounds is also presented in Table 3. Compounds **XII** and **XI** effectively inhibited the activity of the enzyme, as indicated by their IC_50_ values, which were recorded as 92 and 98 μg/mL, respectively. For acarbose (positive control), IC_50_ of 75 μg/mL was noted against α-amylase.

### 2.5. Molecular Docking Studies

The docking method was validated by re-docking of co-crystallized ligands, galantamine, and benzyl pyridinium-4-methyltrichloroacetimidate in the active sites of acetylcholinesterase and butyrylcholinesterase, respectively. The galantamine and benzyl pyridinium-4-methyltrichloroacetimidate were docked efficiently at their binding sites with RMSD (Root Mean Square Deviation) values of 0.88 Å and 1.10 Å in AChE and BChE, respectively. The RMSD value ≤ 3.0 Å is considered significant in re-docking experiments, therefore the docking parameters can predict the binding modes of compounds accurately.

All the compounds (**IX**, **X**, **XI**, and **XII**) were docked in the active sites of AChE and BChE. The most active compound, **XII**, interacted with Tyr121 and surrounding water molecules in the active site of AChE. One of the butanoic acid groups formed a strong hydrogen bond with the side chain -OH of Tyr121 at a distance of 1.68 Å, and a water molecule (1.83 Å). while the other butanoic acid moiety mediated a hydrogen bond with a water molecule (1.90 Å) and π–H interaction with the side chain of Tyr84 (2.70 Å). The docking score of the compound was −6.71 Kcal/mol. Similarly, in the active site of BchE, the compound **XII** interacted with Tyr332. The main chain carbonyl group of Tyr332 accepted a hydrogen bond with one of the butanoic acids of the compound (2.39 Å). Additionally, the phenyl ring of Tyr332 provides π–H interaction to the compound. In the binding site of BChE, the compound exhibited a docking score of −6.65Kcal/mol and did not show any binding with water molecules. The docking scores of galanthamine with AChE and BchE were −9.91Kcal/mol and −7.42 Kcal/mol, respectively, which is higher than the docking scores of compound **XII**. The docking scores of compound **XII** are in good agreement with the IC_50_ values of the compound against AChE and BChE. The binding mode of **XII** in the active site of AChE and BChE is shown in Figure 2.

The docked view of compound **XI** (which is the second most active compound) showed that the dihydroxy moiety of the compounds mediated a strong hydrogen bond with the carbonyl oxygen of His440 of AChE at a distance of 2.22 Å, whereas in the active site of BchE, the side chain of Glu197 interacted with one of the -OH group of the compound via H-bond (bond length = 2.36Å). The docking score of the compound was −4.17Kcal/mol and −4.35 Kcal/mol for AChE and BChE, respectively. The -OH group of compound **IX** donated a hydrogen bond to the backbone carbonyl oxygen of His440 (1.93 Å) in AChE, similarly, the -OH group of **IX** formed H-bond with a water molecule (1.72 Å) in the active site of BChE. Additionally, the side chain of Trp82 of BChE formed π–H interaction with the compound. The docking score of **IX** in the active site of AChE and BChE was −4.01 Kcal/mol and −4.25 kcal/mol, respectively, which is less than the docking scores of compounds **XII** and **XI**. The docking studies further confirm our experimental findings. 

The carboxyl group of the least active molecule, **X** was bound with the side chain of Glu199 via H-bond (1.80 Å), while the side chains of Trp84 and Phe330 stabilize the compounds in the active site of AChE through π–H interaction. Similarly, the carboxyl group of the compounds mediated H-bond with the side chain of Glu197 in the binding site of BChE with a bond length of 1.97 Å, whereas Trp82 formed π–H interaction. The docking score of **X** was −2.87 Kcal/mol and −3.18Kcal/mol, respectively, for AChE and BChE. The docking scores of all the docked compounds correlated well with the IC_50_ of these compounds.

## 3. Discussion

A number of studies suggest a strong pathophysiological link between type 2 diabetes mellitus (T2DM) and AD; diabetic patients are prone to develop AD in their older age. Reactive oxygen species’ toxicity is considered as a strong link between the two diseases and induces their worsening [22,23]. In third world countries, mostly, the peoples rely on plant medications. There is a need to update knowledge about plants’ biological potentials, and the use of scientific approaches in this regard is more useful to evaluate their biological potentials and toxicities properly. Almost all plants contain phenolic compounds, and due to the resonance effect, they can effectively scavenge the free radicals [7,8,9]. As mentioned in the Introduction Section, of the drugs used for the treatment of AD, two are plant products [7,8,9]. Additionally, a number of phytochemicals have been reported to have antidiabetic potentials [10,11,12,13,14,15,16].

In this connection, the present study was designed to evaluate the effectivity of the compounds against those enzymes the inhibition of which could help in T2DM (alpha-amylase and alpha-glucosidase) and AD (cholinesterases), on one hand, and to have free radical scavenging activity, on the other. The purpose was to discover these compound/s that may have all of the three potentials to a significant degree. The isolated compounds in general exhibited good antioxidant potential with the highest free radical scavenging activity shown by compound **XII**. These compounds, specifically compound **XII**, may be used as an antioxidant and would be potentially more helpful if used in combo with naturally occurring antioxidants for improvement of conditions where tissue protection is direly needed. A much-needed and advanced strategy for the treatment of AD is the use of these drugs capable of inhibiting the activity of both AChE and BChE. Researchers are searching for potential inhibitors of both enzymes from plant sources that could be both safe and more effective than the currently available medicines that can only inhibit AChE. Of the isolated compounds, compounds **XII** and **XI** effectively inhibited both of the cholinesterases; the compounds after further in vivo and toxicity studies could be considered as potential therapeutics for the treatment of AD. Likewise, inhibition of the alpha-glucosidase and alpha-amylase could delay the release of free glucose in the blood, and hyperglycemia associated with type 2 can be managed in this way. The compound **XII** was found to be a potent inhibitor of the α-amylase and α-glucosidase. The potential hypoglycemic effect is also supported by the fact that the plant is used as an anti-diabetic remedy in folk medicine. Molecular docking studies of the compounds also support the results obtained for inhibition of the enzymes’ activity. The in vitro potentials of the compounds especially compound **XII** to treat AD (in terms of AChE and BChE inhibition), diabetes (in terms of α-amylase and α-glucosidase inhibition), and oxidative stress (in terms of free radical scavenging) is highly encouraging as the three activities in the same agent is direly needed.

## 4. Materials and Methods

Chromatography was conducted with silica gel 60, mesh size 70–230. Preparative TLC (thin layer chromatography) employed silica gel 60 PF254 (Merck, Darmstadt, Germany). Detection of compounds was performed using UV light at 254 and 266 nm with additional visualization by exposure to iodine vapor and CeSO_4_ spray. ^1^H and ^13^C NMR, COSY (correlated spectroscopy), DEPT (distortionless enhancement by polarization transfer), HMBC, and HSQC (heteronuclear single quantum coherence) spectra were recorded by a Bruker Spectrometer at 600 MHz (Bruker, Billerica, MA, USA), chemical shifts (δ) in ppm, coupling constants (*J*) in Hz were measured.

### 4.1. Plant Collection and Identification

The stem of *Grewia optiva* was collected and identified according to the guidelines of the herbarium at the University of Malakand (UOM/HU/Eth/Collect.0321). After identification of the plant by a taxonomist at the herbarium, a voucher specimen no:1022HU was deposited there.

### 4.2. Extraction and Fractionation

The fresh air-dried stems of *G. optiva* were washed with fresh water to remove dust and other soil constituents. The washed stems were shade dried and then mechanically ground into a fine powder. The powder was then soaked in 95% methanol for one week at room temperature. After filtration, a rotary evaporator (Büchi Rotavapor R-200) was used to concentrate the methanol filtrate under reduced pressure at 40 °C. The semisolid mass obtained was then air dried. The crude extract (510 g) obtained was then fractionated into aqueous, chloroform, ethyl acetate, and petroleum spirit fractions. The fractions were dried using a rotary evaporator at 40–45 °C. After HPLC (high performance liquid chromatography) profiling, two of the fractions (ethyl acetate and chloroform) were put through a silica gel column. Multiple fractions were eluted and based on TLC profiling, some of these were recombined to obtain a final separation of twenty fractions (FA1-FA20).

The FA-9 fraction obtained from the chloroform extract was further sub-fractionated by column purification, using elution with petroleum spirit- CHCl_3_ (7:3). The C-7 subfraction was then further purified on a silica column with petroleum spirit-CHCl_3_ (4:6) elution, resulting in the isolation of compound **IX** (31 mg). The FA-3 fraction obtained from the ethyl acetate extract resulted in the isolation of compound **X** (16 mg) after passing through a pencil column eluted with petroleum spirit-ethyl acetate (5:5). Compound **XI** was isolated from the FA-2 subfraction of ethyl acetate fraction, after passing with silica column eluted with petroleum spirit-ethyl acetate (8:2). Compound **XII** was isolated from A-17 subfraction of chloroform main fraction with further purification on pencil column using petroleum spirit-CHCl_3_ (7:3) for elution.

### 4.3. Determination of Free Radical Scavenging Activities

The free radical scavenging capacity of the isolated compounds was determined using DPPH and ABTS assays [23,24]. To prepare the DPPH stock solution, 30 mg of DPPH was dissolved in 100 mL of methanol. Likewise, to prepare ABTS stock solution, 383 mg of 7 mM ABTS and 66.2 mg of 2.45 mM K_2_S_2_O_8_ were taken, each of them was individually dissolved in 100 mL of methanol and mixed together thoroughly. For the free radical development, the solutions were kept in dark for one day. Different dilutions (62.5, 125, 250, 500, 1000 μg/mL) from each of the sample stock solutions (5 mg/5 mL of methanol) were prepared. Each of the dilutions (0.8 mL) was mixed with 0.8 mL of DPPH stock solution and incubated for 15 min and their absorbance was recorded at 515 nm. Likewise, 0.8 mL of each of the same dilutions was mixed with 2 mL of ABTS solution, incubated for 25 min and their absorbance was recorded at 745 nm. The percent scavenging potentials of compounds were calculated using the following equation:(1)$$\mathrm{Percent}\;\mathrm{scavenging}=\;\frac{A-B}{A}\;\times \;100,\;$$
where A is the absorbance of oxidized stock (DPPH/ABTS) used as control, and B is the absorbance of the sample.

### 4.4. Determination of Anticholinesterase Activities

AChE and BChE were used as representative enzymes to determine the anticholinesterase effect of isolated compounds using Ellman’s assay with slight modifications [21]. Sample stocks (5 mg/5 mL in methanol) were prepared and a range of serial dilutions (125, 250, 500, 1000 μg/mL) was made from each of them. Phosphate buffer (6.8 pH) was used for making the enzyme and substrate solutions. The principle underlying the standard protocol is based on the production of 5-thio-2-nitrobenzoate anion from DTNB (5,5-dithio-bis-(2-nitrobenzoic acid), while the hydrolysis of substrates results in the formation of acetyl and butyryl thiocholine iodide by AChE and BChE, respectively. The resultant anion and enzymatic hydrolysis products then combined, resulting in a pale-yellow color complex. The absorbance of the colored complex was recorded after 15 min incubation through a UV spectrophotometer. Equations (1)–(4) were used to calculate % inhibition and activity.
(2)$$V=\frac{\Delta \mathrm{Abs}}{\Delta T}$$
(3)$$\%\;\mathrm{Enzyme}\;\mathrm{activity}=\;\frac{V}{\mathrm{Vmax}}\;\times \;100$$
(4)% enzyme inhibition = 100 −% enzyme activity,
where V = velocity of the reaction in the presence of each sample, Abs= absorbance while V_max_ is without the presence of any sample.

### 4.5. Effect on the Activity of α-Glucosidase 

α-Glucosidase (0.5 unit/mL) was mixed with phosphate buffer (0.1 M, pH 6.9) to make enzyme stock solution. Likewise, a substrate solution (p-nitrophenyl-α-D-glucopyranoside) was also prepared in phosphate buffer. Each of the sample dilutions was mixed with 1 mL of the enzyme solution separately and incubated at 37 °C for 15 min. Sodium carbonate solution (80 µL of 0.2 M) was added to the reaction mixture to stop the reaction, and the absorbance was measured at 405 nm. The % enzyme inhibition was calculated as
(5)$$\%\;\mathrm{inhibition}=\frac{\mathrm{Control}\;\mathrm{absorbance}-\mathrm{Test}\;\mathrm{absorbance}\;}{\mathrm{Control}\;\mathrm{absorbance}}\;\times \;100$$

### 4.6. Effect on the Activity of α-Amylase

The degree of inhibition was assessed by quantifying maltose equivalents using a slightly modified dinitrosalicylic acid (DNS) method [23]. From each of the sample dilutions, 1 mL was pre-incubated with 1 U/mL α-amylase for half an hour. After incubation, 1 mL of 1% of starch solution was added to the mixture and further incubated for 10 min. To stop the reaction, 1 mL DNS reagent was added to the reaction mixture and heated in a boiling water bath for 5 min. A solution without amylase was used as blank and acarbose as the positive control. The absorbance was measured at 540 nm. The % enzyme inhibition was calculated as follows:(6)$$\%\;\mathrm{reaction}=\frac{\;\mathrm{test}}{\mathrm{control}\;}\;\times \;100$$
(7)% inhibition=100% reaction 

### 4.7. Molecular Docking Simulations

The molecular docking was performed on the three-dimensional (3D-) structures of AChE and BChE. The X-ray crystal structures of AChE (PDB: 1QTI, resolution = 2.50 Å) and BChE (PDB: 4B0O, resolution = 2.35 Å) were taken from the RCSB Protein Data Bank (RCSB PDB) in complex with galanthamine and benzyl pyridinium-4-methyltrichloroacetimidate, respectively. The docking was carried out on Molecular Operating Environment (MOE, version 2014.09). The 2D-structures of ligands were drawn on ChemDraw and optimized on MOE by minimizing each structure using Amber12:EHT force field until the RMS gradient of 0.1 Kcal/mol/Å was obtained. The protein files were prepared by adding hydrogen atoms and partial charges of proteins using Amber12:EHT force field on MOE. Previously, we studied the role of water molecules in the active site of AChE in detail [7,8,9], and therefore, water molecules within 3 Å vicinity of the active site of AChE and BChE were retained in the protein file, while the rest were removed. Moreover, heteroatoms other than co-crystallized ligands (galanthamine in AChE and benzyl pyridinium-4-methyltrichloroacetimidate in BChE) were also removed from protein files. Docking was performed with the Triangle matcher docking algorithm and London dG scoring function of MOE [11].

## 5. Conclusions

In the current study, four bioactive compounds were isolated for the very first time from the stems of *G. optiva*. These isolated compounds were evaluated for antioxidant, antidiabetic, and anticholinesterase activity. The results of the study revealed that compound **XII,** 2,2’-(1,4-phenylene)bis(3-methylbutanoic acid, has anticholinesterase, antidiabetic and antioxidant potentials. Although the potency of this compound is lower than the standard drugs used for comparison in the different assays, its activity against multiple targets (enzyme inhibition and free radical scavenging capacity) may be of potential pharmaceutical benefit. Computational docking studies support the potential use of compound **XII** as an inhibitor of cholinesterases, and therefore a modifying agent in AD. However, further in vivo studies are required to support this connection.

## Figures and Tables

**Figure 1 molecules-26-02019-f001:**
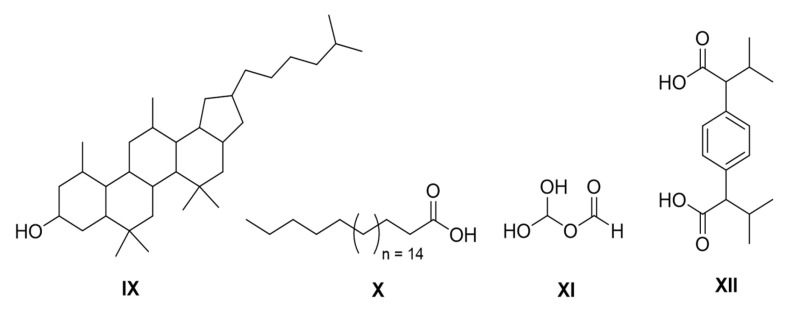
Structure of compounds **IX–XII.**

**Figure 2 molecules-26-02019-f002:**
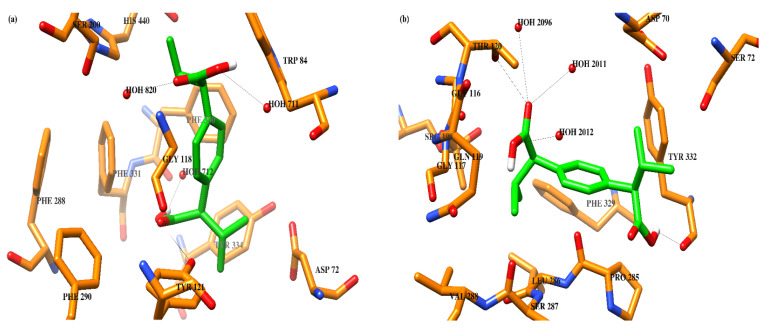
The binding mode of the most active compound **XII** (green) is shown in the active sites of (**a**) AChE (1QTI) and (**b**) BChE (4B0O). The active site residues are shown in the orange stick model, H-bonds are depicted in black lines.

**Table 1 molecules-26-02019-t001:** Percent 2,2-diphenyl-1-picrylhydrazyl (DPPH) and 2,2’-Azino-bis(3-ethylbenzthiazoline-6-sulfonic acid) (ABTS) radical scavenging potential of compounds.

Compounds	Concentrations (μg/mL)	DPPH Percent Inhibition(Mean ± S.E.M)	DPPH IC_50_ (μg/mL)	ABTS Percent Inhibition(Mean ± S.E.M)	ABTS IC_50_ (μg/mL)
**IX**	1000	80.21 ± 040 ^ns^	76	79.23 ± 1.01 **	76
	500	72.19 ± 2.17 **		71.29 ± 1.11 **	
	250	64.11 ± 2.91 **		62.24 ± 2.30 ***	
	125	57.29. ± 2.10 ***		57.19 ± 2.01 **	
	62.5	47.85 ± 2.61 ***		47.83 ± 2.26 **	
**X**	1000	78.81 ± 2.10 **	95	78.03 ± 1.51 **	95
	500	69.19 ± 2.27 **		68.99 ± 1.88 **	
	250	60.61 ± 0.71 **		60.11 ± 1.90 ***	
	125	54.29. ± 2.32 ***		54.13 ± 0.31 **	
	62.5	46.17 ± 3.01 ***		46.01 ± 1.06 **	
**XI**	1000	80.41 ± 2.16 ***	95	79.17 ± 0.53 ***	95
	500	71.17 ± 2.14 **		70.15 ± 1.16 ***	
	250	62.13 ± 1.08 ***		61.04 ± 2.38 ***	
	125	54.15 ± 1.28 ***		54.30 ± 1.15 **	
	62.5	46.22 ± 0.60 ***		46.23 ± 1.49 ***	
**XII**	1000	86.54 ±2.16 ***	60	86.07 ± 1.43 ***	62
	500	78.17 ± 1.04 **		77.15 ± 1.06 ***	
	250	71.13 ± 1.31 ***		71.44 ± 2.11 ***	
	125	61.75 ± 1.78 ***		60.20 ± 3.05 ***	
	62.5	51.02 ± 1.30 ***		50.51 ± 1.19 **	
**Ascorbic Acid**	1000	94.14 ± 1.76	35	93.07 ± 0.53	35
	500	87.87 ± 1.64		83.45 ± 2.26	
	250	78.63 ± 1.48		75.14 ± 3.16	
	125	65.35 ± 1.08		63.30 ± 2.75	
	62.5	55.12 ± 1.30		55.41 ± 1.39	

Ascorbic acid was used a positive control. Data are represented as (mean ± S.E.M) *n* = 3. Values significantly different as compared to positive control, **: *p* < 0.01, ***: *p* < 0.001, ns: *p* > 0.05. IC_50_ = Half-maximal inhibitory concentration while S.E.M = Standard error of the mean.

**Table 2 molecules-26-02019-t002:** Acetylcholinesterase (AChE) and butyrylcholinesterase (BChE) %inhibition exhibited by isolated compounds.

Compounds	Concentrations (μg/mL)	Percent AChE (Mean ± SEM)	AChE IC_50_ (μg/mL)	Percent BChE (Mean ± SEM)	BChE IC_50_ (μg/mL)
**IX**	1000	78.11 ± 0.61 **	90	78.03 ± 2.62 **	90
	500	71.29 ± 1.37 ^ns^		70.19 ± 0.71 **	
	250	63.14 ± 1.51 ^ns^		62.97 ± 1.45 ***	
	125	56.19. ±1.11 ***		55.39 ± 1.01 **	
	62.5	45.95 ± 1.71 ***		45.93 ± 1.26 **	
**X**	1000	70.11 ± 1.40 **	130	69.13 ± 2.01 **	130
	500	62.49 ± 1.37 **		61.97 ± 0.78 **	
	250	56.11 ± 1.51 **		55.51 ± 0.80 ***	
	125	49.29 ± 2.32 ***		49.13 ± 0.31 **	
	62.5	43.17 ± 3.01 ***		43.01 ± 1.06 **	
**XI**	1000	81.41 ± 2.51 **	75	81.31 ± 1.41 **	75
	500	71.19 ± 2.42 **		71.23 ± 1.31 **	
	250	64.51 ± 3.04 **		63.73 ± 2.24 ***	
	125	58.71. ± 2.00 ***		57.15 ± 1.31 **	
	62.5	48.20 ± 3.34 ***		48.02 ± 2.06 **	
**XII**	1000	84.41 ± 1.36 ***	55	93.89 ± 0.81 ***	60
	500	76.16 ± 1.01 ***		85.14 ± 2.06 ***	
	250	69.33 ± 1.38 ***		77.08 ± 2.26 ***	
	125	61.25 ± 1.71 ***		64.42 ± 1.85 ***	
	62.5	52.45 ± 1.47 ***		50.43 ± 1.11 ***	
**Galanthamine**	1000	94.11 ± 1.56	40	93.89 ± 0.81	40
	500	86.17 ± 1.50		85.14 ± 2.06	
	250	77.13 ± 1.08		77.08 ± 2.26	
	125	64.15 ± 2.70		64.42 ± 1.85	
	62.5	94.11 ± 1.56		93.89 ± 0.81	

Galantamine was used a positive control. Data is represented as (mean ± S.E.M) *n* = 3. Values significantly different as compared to positive control, *: *p* < 0.05, **: *p* < 0.01, ***: *p* < 0.001, ns: *p* > 0.05.

**Table 3 molecules-26-02019-t003:** α-glucosidase and α-amylase inhibition of the isolated compounds.

Compounds	Concentrations (μg/mL)	Percent Inhibition of α-Glucosidase (Mean ± S.E.M)	IC_50_ (μg/mL)	Percent Inhibition of α-Amylase (Mean ± S.E.M)	IC_50_ (μg/mL)
**IX**	1000	71.11 ± 1.41 ***	120	70.21 ± 1.91 ***	122
	500	66.19 ± 1.21 ***		65.13 ± 1.43 ***	
	250	60.11 ± 3.23 ***		59.14 ± 0.82 ***	
	125	51.71 ± 1.91 ***		50.61 ± 1.52 ***	
	62.5	41.75 ± 1.61 ^ns^		40.13 ± 1.27 **	
**X**	1000	70.29 ± 1.13 ***	125	70.29 ± 1.13 ***	125
	500	65.19 ± 1.28 ***		65.19 ± 1.28 ***	
	250	59.11 ± 2.13 ****		59.11 ± 2.13 ***	
	125	50.11 ± 1.31 ***		50.11 ± 1.31 ***	
	62.5	40.70 ± 1.01 ***		40.01 ± 2.06 ^ns^	
**XI**	1000	82.29 ± 1.13 ***	100	81.90 ± 0.73 ***	98
	500	72.21 ± 1.21 ***		70.91 ± 2.11 ***	
	250	60.11 ± 1.82 ***		63.21 ± 1.69 ***	
	125	53.73 ± 1.31 ***		54.13 ± 2.17 ***	
	62.5	42.12 ± 1.63 ***		46.98 ± 2.49 ***	
**XII**	1000	81.14 ± 1.06 ***	90	79.07 ± 1.43 ***	92
	500	75.11 ± 2.04 ***		74.25 ± 1.06 ***	
	250	69.13 ± 2.01 ***		66.94 ± 1.11 ***	
	125	58.05 ± 1.78 ***		56.10 ± 2.05 ***	
	62.5	46.02 ± 1.30 ***		47.51 ± 1.19 ***	
**Acarbose**	1000	93.72 ± 1.03	70	92.12 ± 0.83	75
	500	78.61 ± 1.10		77.11 ± 1.41	
	250	67.16 ± 1.58		66.37 ± 1.18	
	125	58.73 ± 1.09		57.03 ± 1.19	
	62.5	48.76 ± 2.20		48.13 ± 2.09	

Acarbose was used a positive control. Data are represented as (mean ± S.E.M) *n* = 3. Values significantly different as compared to positive control, *: *p* < 0.05, **: *p* < 0.01, ***: *p* < 0.001, ns: *p* > 0.05.

## Data Availability

The data are available to the researchers upon request.

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
