# Peer review of "Bio-Potency and Molecular Docking Studies of Isolated Compounds from Grewia optiva J.R. Drumm. ex Burret"

_molecules, 2021, doi:10.3390/molecules26072019_

Round 1

Reviewer 1 Report

Dear authors,

I have read very carefully your manuscript and I have found some issues regarding the validation and representation of that work.

  1. The introduction is too short and does not provide enough information about previously reported studies in this area nor the importance of your work. More scientific literature must be included to provide valid information about the nature of the study.
  2. The molecular docking studies do not seem to be validated in any way. Please find an appropriate way to validate your results.
  3. Figures 1 and 2 are in poor resolution and must be created again with a better one.
  4. In the materials and methods section, you said that you have run HSQC, HMBC, COSY and DEPT NMR experiments. Despite that, I haven't seen any 2D NMR spectrum in the main manuscript and in supplementary materials as well except one HMBC. 2D NMR is very important in order to be able to characterize adequately plant extracts. So, please provide the 2D NMR spectrums in the main manuscript in a publishable way, providing information about the extract characterization.

Best regards

Author Response

Reviewer 1

Dear authors,

I have read very carefully your manuscript and I have found some issues regarding the validation and representation of that work.

  1. The introduction is too short and does not provide enough information about previously reported studies in this area nor the importance of your work. More scientific literature must be included to provide valid information about the nature of the study.
  • Worthy reviewer, the introduction section has been revised accordingly and hopefully it will be ok now.
  1. The molecular docking studies do not seem to be validated in any way. Please find an appropriate way to validate your results.
  • The molecular docking protocol was validated by redocking of co-crystallized ligands. We have incorporated the redocking results in the revised manuscript.
  1. Figures 1 and 2 are in poor resolution and must be created again with a better one.
  • Now we have removed previous figures, and new figure with high resolution is added in the revised manuscript
  1. In the materials and methods section, you said that you have run HSQC, HMBC, COSY and DEPT NMR experiments. Despite that, I haven't seen any 2D NMR spectrum in the main manuscript and in supplementary materials as well except one HMBC. 2D NMR is very important in order to be able to characterize adequately plant extracts. So, please provide the 2D NMR spectrums in the main manuscript in a publishable way, providing information about the extract characterization.
  • Response: We have incorporated the 2D required NMR techniques for new compound XII which confirm the structure of the compound. Regarding, compound XI is very small molecule having only one aldehyde and one anomeric proton. As there is no stereochemistry involved and no proton-proton correlation was observed in the molecule that’s why only HMBC was run to confirm the position of functional groups. In addition, we removed the words COSY, and HSQC from the material and methods section.

Reviewer 2 Report

Manuscript ID - molecules-1147578

Bio-potency and molecular docking studies of isolated compounds from Grewia optiva J.R. Drumm. ex Burret 3

Wasim Ul Bari1, Najeeb Ur Rehman2, Ajmal Khan2, Ye Yuan3, Mark A. T. Blaskovich, Zyta M. Ziora3, Muhammad 4, Zahoor i, 4, Sumaira Naz 4, Riaz Ullah5, Siddique A. Ansari6, Hafiz Majid Mahmood7 and Ahmed bari8

Review

The authors analyzed four new bioactive chemical compounds isolated from Grewia optiva. They found compound XII to have antioxidant, antidiabetic and anticholinesterase activity. The authors have published in 2019. and 2020. at least three other articles on a similar topic, two of which have not been cited in this manuscript, and one has been cited in the section Material and methods. Although the study is interesting, it cannot be overlooked that the basic concept of the manuscript with the different chemical components, was repeated several times in different journals. In addition, a number of shortcomings have been noted in this manuscript, which is why major revision is recommended.

Major remarks:

Abstract: Authors claim that they identified two new and two novel compounds in extract of G. optiva. Please, clearly state which components are new and which are novel.

The introduction does not provide sufficient background and does not include all relevant references. If one type ‘Grewia optiva’ into Google Schoolar, several thousand of different entries are shown, many of which are scientific articles, that report results of different studies of biological effects of extracts of G. optiva.

Therefore, I recommend to include to this section more articles dealing with bio-potency of G. optiva and three important publications which come from the same (or similar) group of authors, and have a quite similar overall study design:

  1. Bari, W.U. Anticholinesterase, antioxidant potentials, and molecular docking studies of isolated bioactive compounds 364 from Grewia optiva. International Journal of Food Properties 2019, 22, 1386–1396. (This study has been cited but only in terms of the method used (line 248. 4.4. Determination of anticholinesterase activities).
  2. Ul Bari W, Zahoor M, Zeb A, et al. Isolation, pharmacological evaluation and molecular docking studies of bioactive compounds from Grewia optiva. Drug Des Devel Ther. 2019;13:3029-3036. Published 2019 Aug 26. doi:10.2147/DDDT.S220510
  3. Muhammad Zahoor, Wasim Ul Bari, Alam Zeb and Irfan Khan: Toxicological, anticholinesterase, antilipidemic, antidiabetic and antioxidant potentials of Grewia optiva Drummond ex Burret extracts, 2020, Journal of Basic and Clinical Physiology and Pharmacology. Volume 31 Issue 2, DOI: https://doi.org/10.1515/jbcpp-2019-0220

Results: The results of spectroscopic analysis (lines 190 – 228), along with the Figure 2 should be transferred to the section Results, as the first subsection, 2.1. They do not belong to the section Material and methods. The quality of the Figures 1 and 2 is not satisfactory. A better quality image is needed. Also the legend should be clearer with more details. What do the labels a, b, c, d mean in the Fig. 1?  I recommend authors to include the name of the software used or to quote an article or web page.

Discussion – only one article has been mentioned (no 25). Discussion needs to be thoroughly rewritten, numerous articles have been published on the topic of biological activity and chemical composition of extract of Grewia optiva. It is necessary to increase the number of cited articles and to compare the results obtained in this study with results from other studies.

Minor remarks:

Key words: Please add: anticholinesterase activity, chemical composition  

Line 4.  Mark. A.T. Blaskovich – there is no affiliation, please insert no 3 as superscript

Line 5. Ahmed bari, - please replace small with capital letter (Bari)

Line 59: Please write full name of Alzheimer disease not only abbreviation AD

Line 79: Please, use full names with abbreviations in parentheses.

Line 83: Use small leter for cholinesterases

Line 91: Please, use full names with abbreviations in parentheses.

Lines 309-311: Author’s contribution: There are 11 co-authors, and the author’s contribution is given only for 7 authors. Contribution of 4 authors is unclear. Please write contribution of all authors.

Lines 318-319 . Please add the name of the journal.

Author Response

Reviewer 2

The authors analyzed four new bioactive chemical compounds isolated from Grewia optiva. They found compound XII to have antioxidant, antidiabetic and anticholinesterase activity. The authors have published in 2019. and 2020. at least three other articles on a similar topic, two of which have not been cited in this manuscript, and one has been cited in the section Material and methods. Although the study is interesting, it cannot be overlooked that the basic concept of the manuscript with the different chemical components, was repeated several times in different journals. In addition, a number of shortcomings have been noted in this manuscript, which is why major revision is recommended.

  •  

Major remarks:

Abstract: Authors claim that they identified two new and two novel compounds in extract of G. optiva. Please, clearly state which components are new and which are novel.

Response: The compounds are new, not novel. Compounds IX and X are known, while compounds XI and XII are new. We have corrected in the revised version.

The introduction does not provide sufficient background and does not include all relevant references. If one type ‘Grewia optiva’ into Google Schoolar, several thousand of different entries are shown, many of which are scientific articles, that report results of different studies of biological effects of extracts of G. optiva.

Therefore, I recommend to include to this section more articles dealing with bio-potency of G. optiva and three important publications which come from the same (or similar) group of authors, and have a quite similar overall study design:

  1. Bari, W.U. Anticholinesterase, antioxidant potentials, and molecular docking studies of isolated bioactive compounds from Grewia optiva. International Journal of Food Properties 2019, 22, 1386–1396. (This study has been cited but only in terms of the method used (line 248. 4.4. Determination of anticholinesterase activities).
  2. Ul Bari W, Zahoor M, Zeb A, et al. Isolation, pharmacological evaluation and molecular docking studies of bioactive compounds from Grewia optiva. Drug Des Devel Ther. 2019;13:3029-3036. Published 2019 Aug 26. doi:10.2147/DDDT.S220510
  3. Muhammad Zahoor, Wasim Ul Bari, Alam Zeb and Irfan Khan: Toxicological, anticholinesterase, antilipidemic, antidiabetic and antioxidant potentials of Grewia optiva Drummond ex Burret extracts, 2020, Journal of Basic and Clinical Physiology and Pharmacology. Volume 31 Issue 2, DOI: https://doi.org/10.1515/jbcpp-2019-0220
  • Worthy reviewer, the introduction was updated accordingly and suggested references were added accordingly.

Results: The results of spectroscopic analysis (lines 190 – 228), along with the Figure 2 should be transferred to the section Results, as the first subsection, 2.1. They do not belong to the section Material and methods. The quality of the Figures 1 and 2 is not satisfactory. A better quality image is needed. Also the legend should be clearer with more details. What do the labels a, b, c, d mean in the Fig. 1?  I recommend authors to include the name of the software used or to quote an article or web page.

  • Worthy reviewer, the spectral data was moved to result section and placed in sub heading 2.1. The quality of the figure 1 and were improved accordingly. However, as per your suggestion the spectral data has been moved to result section that is why figure 2 is now figure 1 while figure 1 is 2.

Discussion – only one article has been mentioned (no 25). Discussion needs to be thoroughly rewritten, numerous articles have been published on the topic of biological activity and chemical composition of extract of Grewia optiva. It is necessary to increase the number of cited articles and to compare the results obtained in this study with results from other studies.

  • Worthy reviewer, discussion has been improved and new references were added accordingly.

Minor remarks:

Key words: Please add: anticholinesterase activity, chemical composition  

  • They were inserted accordingly, thank you worthy reviewer for your valuable guidance.

Line 4.  Mark. A.T. Blaskovich – there is no affiliation, please insert no 3 as superscript

  • The affiliation was inserted accordingly.

Line 5. Ahmed bari, - please replace small with capital letter (Bari)

  • Worthy reviewer, thank you for pointing out the mistake, it was corrected accordingly.

Line 59: Please write full name of Alzheimer disease not only abbreviation AD

  • Worthy reviewer suggestion was endorsed accordingly.

Line 79: Please, use full names with abbreviations in parentheses.

  • Worthy reviewer, thank you they were defined in parenthesis.

Line 83: Use small leter for cholinesterases

  • Thank you worthy reviewer, they were corrected accordingly.

Line 91: Please, use full names with abbreviations in parentheses.

  • Worthy reviewer, they were accordingly defined in line 85 and 87.

Lines 309-311: Author’s contribution: There are 11 co-authors, and the author’s contribution is given only for 7 authors. Contribution of 4 authors is unclear. Please write contribution of all authors.

  • Worthy reviewer, their contribution was added accordingly.

Lines 318-319 . Please add the name of the journal.

  • Thank you worthy reviewer, it was inserted accordingly.

Round 2

Reviewer 1 Report

The manuscript has been improved significantly and I feel that is ready for publication as it is.

Reviewer 2 Report

The manuscript has been significantly improved so I recommend accepting it. However, I found a few minor language errors, so I recommend an additional check of the English language.

Please correct:

Line 33. 'by them’ to ‘by their’

Line 267: ‘due resonance effect’ please change to ‘due to resonance effect’

Lines 401-402: ‘Previously, we have studied the role of water molecules in the active site of AChE in detail (ref)’ – please cite the reference